# Comparative Metabolic Study of *Tamarindus indica* L.’s Various Organs Based on GC/MS Analysis, *In Silico* and *In Vitro* Anti-Inflammatory and Wound Healing Activities

**DOI:** 10.3390/plants12010087

**Published:** 2022-12-23

**Authors:** Shaza H. Aly, Mahmoud A. El-Hassab, Sameh S. Elhady, Haidy A. Gad

**Affiliations:** 1Department of Pharmacognosy, Faculty of Pharmacy, Badr University in Cairo, Cairo 11829, Egypt; 2Department of Medicinal Chemistry, Faculty of Pharmacy, King Salman International University (KSIU), South Sinai 46612, Egypt; 3Department of Natural Products, Faculty of Pharmacy, King Abdulaziz University, Jeddah 21589, Saudi Arabia; 4Center for Artificial Intelligence in Precision Medicines, King Abdulaziz University, Jeddah 21589, Saudi Arabia; 5Department of Pharmacognosy, Faculty of Pharmacy, Ain Shams University, Cairo 11566, Egypt; 6Department of Pharmacognosy, Faculty of Pharmacy, King Salman International University (KSIU), South Sinai 46612, Egypt

**Keywords:** *Tamarindus indica* L., GC/MS, anti-inflammatory, wound healing, molecular docking, chemometric analysis, drug discovery, public health

## Abstract

The chemical composition of the *n*-hexane extract of *Tamarindus indica’s* various organs—bark, leaves, seeds, and fruits (TIB, TIL, TIS, TIF)—was investigated using gas chromatography-mass spectrometry (GC/MS) analysis. A total of 113 metabolites were identified, accounting for 93.07, 83.17, 84.05, and 85.08 % of the total identified components in TIB, TIL, TIS, and TIF, respectively. Lupeol was the most predominant component in TIB and TIL, accounting for 23.61 and 22.78%, respectively. However, *n*-Docosanoic acid (10.49%) and methyl tricosanoate (7.09%) were present in a high percentage in TIS. However, *α*-terpinyl acetate (7.36%) and *α*-muurolene (7.52%) were the major components of TIF *n*-hexane extract. By applying a principal component analysis (PCA) and hierarchal cluster analysis (HCA) to GC/MS-based metabolites, a clear differentiation of *Tamarindus indica* organs was achieved. The anti-inflammatory activity was evaluated in vitro on lipopolysaccharide (LPS)-induced RAW 264.7 macrophages. In addition, the wound healing potential for the *n*-hexane extract of various plant organs was assessed using the in-vitro wound scratch assay using Human Skin Fibroblast cells. The tested extracts showed considerable anti-inflammatory and wound-healing activities. At a concentration of 10 µg/mL, TIL showed the highest nitric oxide (NO) inhibition by 53.97 ± 5.89%. Regarding the wound healing potential, after 24 h, TIB, TIL, TIS, and TIF *n*-hexane extracts at 10 g/mL reduced the wound width to 1.09 ± 0.04, 1.12 ± 0.18, 1.09 ± 0.28, and 1.41 ± 0.35 mm, respectively, as compared to the control cells (1.37 ± 0.15 mm). These findings showed that the *n*-hexane extract of *T. indica* enhanced wound healing by promoting fibroblast migration. Additionally, a docking study was conducted to assess the major identified phytoconstituents’ affinity for binding to glycogen synthase kinase 3-β (GSK3-β), matrix metalloproteinases-8 (MMP-8), and nitric oxide synthase (iNOS). Lupeol showed the most favourable binding affinity to GSK3-β and iNOS, equal to −12.5 and −13.7 Kcal/mol, respectively, while methyl tricosanoate showed the highest binding affinity with MMP-8 equal to −13.1 Kcal/mol. Accordingly, the *n*-hexane extract of *T. indica’s* various organs can be considered a good candidate for the management of wound healing and inflammatory conditions.

## 1. Introduction

*Tamarindus* is a monospecific genus belonging to the Fabaceae family and usually has been identified under the name Tamarind [1]. *Tamarindus indica* Linn. is a tropical fruit tree that can be found growing wild in many tropical and subtropical areas [2]. Tamarind is a semi-deciduous tree, slow growing but long-lived, that can reach a height of up to 30 m. The leaves are alternate and paripinnate, and its fruit is a legume appearing as indehiscent pods with 1–10 seeds, which represent the main criteria of plants belonging to the Fabaceae family [3,4,5,6]. The fruit was especially well-known and commonly used by the ancient Egyptians and widely consumed and traded in Africa [3]. The fruit has usually been used in food dishes through different applications as a condiment and spice and could be eaten fresh or raw. Tamarind is sold as an entire pod, paste, or concentrate in speciality food stores all throughout the world [7,8]. Its fruit has been reported in the literature for its pharmacological antioxidant, anti-inflammatory, antidiabetic, anti-Alzheimer, and antihyperlipidemic properties [9,10,11]. Tamarind has been commonly used in folk medicine for the treatment of diarrhoea and dysentery, helminth infections, fever, and abdominal pain. It is also used as an anti-malarial and laxative and has a role in wound healing [3]. Traditional Indian and African medicine has made substantial use of different tamarind products, including its leaves, fruit, and seeds [12]. Across many areas of central West Africa, tamarind leaves and bark are used traditionally to alleviate wounds [3,13]. Leaves of Tamarind have nutritional value due to their richness in vitamins and minerals [12]. Its leaves have exhibited potential antioxidant, antibacterial and antifungal, anti-inflammatory, and analgesic activities [14,15,16,17]. Additionally, the leaves of Tamarind have an anti-inflammatory role in Indian folk medicine [18].

Regarding the bark, it is effective as an astringent and tonic agent in lotions or poultices to relieve sores, ulcers, boils, and rashes, owing to its richness with tannins [12]. In addition, the bark showed in vivo antihyperglycemic, anti-inflammatory, and analgesic activities [17,19]. Moreover, leaves and bark have been reported for their anthelmintic activity [20].

Wound healing is a normal complex repair process that requires synchronisation between various biological and immunological systems, including many phases of inflammation, proliferation, and contraction of the wound, with the formation of granulation tissues and remodelling [21,22,23]. Plants and plant products, such as *Aloe vera*, *Sophora flavescens, Punica granatum* L., *Mimosa pudica* L., and *Hibiscus rosa sinensis* L., have been used in traditional medicine to cure and prevent diseases, especially wounds [24,25,26].

Recently, there has been a great demand for the development of natural products to cure different conditions owing to their safety, availability, versatile biological activities, and unique secondary metabolites [27,28,29,30,31]. Consequently, the exploration of medicinal plants as new candidates for wound healing would be valuable and beneficial, especially medicinal plants that are characterised by biocompatibility, wound healing, and anti-inflammatory properties [22,23,32,33]. Many reports have revealed that using *n*-hexane as an extracting solvent targets non-polar compounds [21,34,35], and many studies have proven the role of *n*-hexane extract from various plants in anti-inflammatory and wound healing conditions [36,37,38,39].

Thus, the purpose of this present study was to compare and determine the chemical composition of the *n*-hexane extract of *T. indica* L.’s various organs using GC/MS analysis. Further, the principal component analysis (PCA) and hierarchical cluster analysis (HCA) were applied to discriminate among various plant organs based on their chemical profiles, which could be used as a tool for the detection of similarities and differences among the four organs. Furthermore, the *n*-hexane extract obtained from *T. indica* L.’s various organs were investigated for their anti-inflammatory and wound healing properties for the first time along with a molecular docking study to correlate the chemical composition with the mentioned biological activity. To the best of our knowledge, this is the first study on the wound-healing properties for the various organs of *T. indica* L. *n*-hexane extract.

## 2. Results and Discussion

### 2.1. GC/MS Analysis of the N-Hexane Extract of Tamarindus indica *L.* Various Organs

The chemical composition of the *n*-hexane extract of *T. indica*, bark, leaves, seeds, and fruit (TIB, TIL, TIS, and TIL) were investigated using GC/MS analyses (Table 1). The yields of extraction of TIB, TIL, TIS, and TIL using *n*-hexane were 4.20, 6.10, 2.07, and 3.12% *w*/*w*, respectively. The samples encompassed 113 constituents that accounted for 93.07, 83.17, 84.05, and 85.08 % of the total identified components in bark, leaves, seeds, and fruits of *T. indica*, respectively. The major constituents identified in the *n*-hexane extract of TIB were lupeol (23.61%), lupeol acetate (23.02%), lupenone (9.36%), *n*-tetratriacontane (8.16%), 24-methylenecycloartanol (5.87%), 1-heptatriacontanol (4.25%), and *γ*-sitosterol (4.17%). Regarding the *n*-hexane extract of TIL, lupeol (22.78%), lupenone (13.08%), *γ*-sitosterol (8.83%), (17*E*)-Cholesta-17,24-diene-3,6-diol (8.23%), *n*-nonacosane (4.13%), Betulinaldehyde (3.70%), and *β*-Amyrone (2.05%) were characterised as the chief metabolites. In this case, lupenone and lupeol were present in higher percentages as compared to other organs. Moreover, *n*-Docosanoic acid (10.49%), methyl tricosanoate (7.09%), tetracosanal (5.42%), 2-methylhexacosane (4.31%), 11-methylpentacosane (4.22%), and heneicosyl acetate (2.94%) displayed the highest percentages in the *n*-hexane extract of TIS. Additionally, among the major constituents, especially in TIS, are *n*-octacosane (4.63%), *n*-heptacosane (4.17%), *cis*-13,16-docosadienoic acid (3.70%), *n*-hexacosane (2.47%), *n*- pentacosane (2.46%), and methyl tetracosanoate (2.17%). Furthermore, the *n*-hexane extract of TIF showed a predominantly high percentage of *α*-terpinyl acetate (7.36%), α-muurolene (7.52%), 2-methyl hexadecane (5.70%), methyl pentadecanoate (4.53%), *α*-eicosene (3.44%), and linolenic acid (3.26%) in addition to chief constituents *n*-heptacosane (6.67%), *n*-nonacosane (5.42%), squalene (4.17%), *cis*-13,16-docosadienoic acid (3.76%), lupeol (3.76%), *n*-tetratriacontane (3.38%), and hentriacontane (3.32%). It was notable that *γ*-sitosterol is the chief component in the four organs ranging from (3.79%) in TIS to (11.28%) in TIF. It was notable that squalene, n-nonacosane, n-hentriacontane, 5*α*-stigmast-22-en-3*β*-ol, and *γ*-sitosterol were detected in the four organs. The major compounds present in the four various organs of *T. indica* are represented in (Figure 1).

Triterpenoids and steroids are the predominant classes in the *T. indica* bark *n*-hexane extract, accounting for 61.06 % and 13.47%, respectively. Additionally, the *n*-hexane extract of *T. indica* leaves showed a high percentage of triterpenoids and steroids, representing 47.41% and 11.86% of the identified compounds, respectively, along with fatty acids and their derivatives accounting for 8.40 %. On the other hand, the most predominant classes of metabolites in the *n*-hexane extract of *T. indica* seeds were fatty acids and their derivatives and the straight-chain hydrocarbons, accounting for 35.22% and 40.36%, respectively. Meanwhile, the straight-chain hydrocarbons were the dominant class in the *n*-hexane extract of *T. indica* fruits, representing 33.55% of the total identified compounds, followed by steroids and fatty acids and their derivatives, accounting for 12.67% and 11.55%, respectively. The illustration of the metabolite distribution in the various organs of *T. indica* is represented in (Figure 2).

The fatty acids previously identified in the seeds of Tamarind from Sudan by Ibrahim et al. [40] were characterised as palmitic acid, oleic acid, and eicosanoic acid in addition to the identification of *β*-amyrin, *β*-sitosterol, and campesterol. *Cis*-vaccenic acid, 2-methyltetracosane, *β*-sitosterol, 9,12-octadecadienoic acid (*Z*, *Z*)-, and n-hexadecanoic acid were reported as the major metabolites identified in the oil of *T. indica* seeds collected from Nigeria. Another study by Carasek and Pawliszyn [41] reported that the volatile compounds in the tamarind fruit collected from Brazil included phenylacetaldehyde and furfural. Aldehyde and ester compounds were mostly identified by using Solid-Phase Microextraction (SPME) Fibres. It is important to note that the plant extract components can be influenced by variables including geographic, climatic, seasonal, and experimental settings. However, the aim of our study was to explore the difference in discriminating among the various organs of *T. indica*, not to study the effects of the above-mentioned variables from Gad et al. [42].

### 2.2. Chemometric Analysis Based on GC/MS Analysis

GC/MS-based metabolites showed both qualitative and quantitative variation among different *Tamarindus indica* organs. Consequently, chemometric analysis, representing both principal component analysis (PCA) and hierarchal cluster analysis (HCA), was applied to the relative peak areas of all the identified compounds of different *T. indica* organs to explore the similarities and differences among them. The PCA score plot and loading plot are shown in Figure 3a,b. In the PCA score plot, the percentage of the first two principal components explained 83% of the variance in the data, where PC1 accounted for 63%, while PC2 explained only 20% of the data discrepancies. From the score plot, various *T. indica* organs were divided into four main groups, and each part was scattered in a separate quadrant. TIL and TIB were positioned on the positive side of PC1, whereas, TIF and TIS were located on the negative side of PC1. An in-depth inspection of the loading plot (Figure 3b) revealed that lupeol acetate was the major discriminating marker for TIB. Although lupeol was the major identified component in both TIB and TIL, it was ©dentified as the main segregating marker for TIL in addition to lupenone.

Regarding TIF and TIS, γ-Sitosterol and n-Docosanoic acid were the chef compounds responsible for their separation in separate quadrants. By applying HCA, the dendrogram (Figure 3c) categorised *T. indica* organs into four main clusters. Cluster I, II, III, and IV displayed TIL, TIB, TIF, and TIS, respectively. The dendrogram showed that TIL and TIB were closely related to each other’s when compared to TIF and TIS. Both PCA and HCA successfully discriminated among various *T. indica* organs proving the effective application of chemometric analysis in combination with GC/MS-based metabolites for the discrimination among various plant organs.

### 2.3. Anti-Inflammatory Activity of Tamarindus indica *L.* Various Organs

The anti-inflammatory effects of the *n*-hexane extract of *T. indica* bark, leaves, seeds, and fruits (TIB, TIL, TIS, and TIF) on lipopolysaccharide (LPS)-induced RAW 264.7 macrophages were represented in (Table 2). The various organs of *T. indica* showed promising inhibition of LPS-induced nitric oxide (NO) release in RAW 264.7 macrophages. At a concentration of 10 µg/mL, TIB, TIL, TIS, and TIF *n*-hexane extracts showed %NO inhibition by 7.53 ± 1.69, 53.97 ± 5.89, 19.54 ± 1.19, and 26.66 ± 3.44%, respectively. Further, at a concentration of 100 µg/mL, TIB, TIL, TIS, and TIF *n*-hexane extracts showed %NO inhibition by 51.44 ± 1.17, 98.00 ± 1.90, 85.47 ± 0.22, and 83.69 ± 2.39%, respectively as compared to positive control L-N^G^-nitro arginine methyl ester (L-NAME) (1 mM) that showed 84.64 ± 1.04% NO inhibition.

Previous studies revealed that the fruit pulp was applied to inflammations and traditionally used as a gargle for sore throats. Further, the bark decoction was used in cases of eye inflammation [12]. Rimbau et al. revealed the in vivo anti-inflammatory properties of different extracts of tamarind fruit pulp in mice ear oedema induced by arachidonic acid, and rats sub plantar oedema induced by carrageenan [43]. Moreover, the petroleum ether and ethyl acetate extracts of seeds of *T. indica* showed a significant anti-inflammatory and analgesic potential using carrageenan-induced paw oedema and cotton pellet-induced granuloma models in rats [44]. Among the major compounds in the *n*-hexane extract of *T. indica’s* various organs were lupenone and sitosterol, which were reported for their anti-inflammatory activity [45]. Additionally, among the major identified compounds, lupeol and lupeol acetate were reported for their anti-inflammatory properties through regulating TNF-α, IL-2, and IL-β specific mRNA, reducing PGE-2 synthesis from macrophage, neutrophil migration, and the number of iNOS cells [46].

### 2.4. Wound Healing Activity of Tamarindus indica *L.* Various Organs

#### 2.4.1. Cytotoxicity Assay

Evaluation of the cytotoxicity of the *n*-hexane extract of *T. indica* bark, leaves, seeds, and fruits (TIB, TIL, TIS, and TIL) on the HSF cells was necessary to assess the safety of the treatment dose. The cytotoxicity on HSF cells of the *n*-hexane extract of *T. indica’s* various organs was evaluated using the SRB assay [47]. In our investigation, TIF and TIS (up to 100 μg/mL) had no cytotoxic effect on the HSF cells. While TIB and TIL at a concentration of 10 μg/mL showed HSF cell viability up to 91.04 ± 2.09 and 95.38 ± 0.86%, respectively (Figure 4). Therefore, a concentration of 10 μg/mL was safe and used further as the treatment dose in the scratch wound assay.

#### 2.4.2. Scratch Wound Assay

In the present study, an in vitro scratch assay using HSF cells was used to assess TIB, TIL, TIS and TIF wound-healing activity. It was evaluated by changes in wound width by measuring the average distance between the borders of the scratches (Figure 5 and Figure 6). The tested plant extracts at a concentration of 10 μg/mL decreased the wound width significantly compared with the control cells (Table 3).

After 24 h, the highest wound healing potential was recorded by TIB and TIS *n*-hexane extracts with wound widths equal to 1.09 ± 0.04 and 1.09 ± 0.28 mm, respectively. This result was followed by TIL and TIF with wound widths equal to 1.12 ± 0.18 and 1.41 ± 0.35 mm, respectively, as compared to the wound width in the control cells that equalled 1.37 ± 0.15 mm. As the wound width reduced as cell migration was enhanced, our findings revealed that both TIB and TIL *n*-hexane extracts exhibited almost complete cell migration after 48 h of observation. Meanwhile, TIS and TIF *n*-hexane extracts had wound-healing effects on HSF cells, but the time required to conduct wound closure was longer than for the negative control. This revealed that the *n*-hexane extract of *T. indica’s* various organs exhibited wound-healing potential through fibroblast migration enhancement.

Previous reports revealed the traditional utilisation of tamarind in eye surgery for conjunctival cell adhesion and corneal wound healing [12]. In a study by Adeniyi et al., the wound-healing activity of *T. indica* leaves and pulp was investigated in the African catfish, and the results revealed the enhancement of the wound healing significantly in the fish-fed tamarind-fortified diets as a consequence of the elevation of antioxidant enzymes [48]. Attah et al. investigated the wound-healing potential of *T. indica* fruit paste in adult rabbits, and the results revealed wound closure acceleration and increasing epithelial migration and re-epithelialisation [49].

The wound-healing potential of *n*-hexane extracts of various organs of *T. indica* can be linked to their chemical components. For example, *γ*-sitosterol is one of the most famous phytosterols with reported biological activity. Phytosterols were shown to inhibit MMP-1, reduce collagen breakdown, and promote the synthesis of collagen in human keratinocytes [50]. Moreover, they promote keratinocyte migration through the reduction of oxidative stress that effectively accelerates the healing process [51]. In previous studies, lupeol and lupeol acetate, which are the major identified components in the present study, were reported for their wound-healing properties [52,53,54,55,56,57,58]. Lupeol prevents collagen I depletion and restores levels of hydroxyproline, hexosamine, hexuronic acid, and matrix glycosaminoglycans, together with modulating collagen I expression in human fibroblasts during the proliferation phase. Further, it enhances fibroblast proliferation, angiogenesis, and growth factors in wound healing [52,53,54]. Another study reported the potential of lupeol for wound healing in streptozotocin-induced hyperglycaemic rats [56]. Another study by Malinowska et al. revealed that lupeol acetate was one of the most effective lupeol derivatives in the stimulation of the human skin cell proliferation process [58]. Bopage et al. reported that the presence of a 3-OH group in the lupeol structure is one of the essential features in the lupane skeleton for wound healing activity as compared to lupenone and lupeol acetate [59].

Linolenic acid and squalene are among the major identified constituents in the *n*-hexane extract of *T. indica* fruits; linolenic acid plays an important role in the wound healing process across its antioxidative and anti-inflammatory properties as well as by encouraging cell proliferation, raising collagen synthesis, promoting dermal reconstruction, and restoring the function of the skin’s lipid barrier [60,61,62,63]. Further, squalene is reported to have anti-inflammatory and protective actions against skin damage. It can accelerate the healing of wounds by stimulating the macrophage response to inflammation. Squalene could be helpful during the resolution phase of wound healing [64].

The amounts of lupeol, lupeol acetate, lupenone, and *γ*-sitosterol show a correlation with the in vitro wound healing potential of the *n*-hexane extract of various organs of *T. indica*, and TIB was found to contain the highest amounts of these compounds according to the GC/MS analysis. The present study and the data in the literature show that terpenoids, sterols, and fatty acids play important roles in the wound-healing potential of *T. indica*. So, *T. indica* is suggested as a potent natural wound-healing product.

### 2.5. In Silico Molecular Docking Studies

This part was conducted to investigate the possible mechanism of action in which the identified major compounds exert their biological effect. Accordingly, the 3D structures of glycogen synthase kinase 3-β (GSK3-β), matrix metalloproteinases-8 (MMP-8), and nitric oxide synthase (iNOS) were downloaded from the protein data bank using the following IDs: 3F88, 5H8X, and 3N2R, respectively. After that, the twenty major compounds were docked into the active site vicinity of the three enzymes. Interestingly, all the compounds achieved acceptable binding scores upon docking with the three targets (Table 4).

To this end, it was expected that the identified major compounds would exert synergetic effects. In the docking of GSK3-*β*-lupeol, n-docosanoic acid, methyl tricosanoate, *α*-terpinyl acetate, and lupeol acetate achieved best docking scores of −12.5, −11.7, −11.2, −11.8, and −11.3 Kcal/Mol, respectively. As Figure 7 reveals, lupeol interacted with Val135, Tyr140, Arg141, Gln185, and Cys199; n-docosanoic acid interacted with Gly63, Lys183, and Cys199; methyl tricosanoate interacted with GSK3-β through binding with Phe67, Arg141, and Cys199; *α*-terpinyl acetate interacted with Lys85, Asp133, and Cys199; and lupeol acetate interacted with Lys85, Gln185, and Cys199. In the docking of matrix metalloproteinases-8 (MMP-8), lupeol, n-docosanoic acid, methyl tricosanoate, *α*-terpinyl acetate, and *α*-muurolene achieved the best docking scores of −9.8, −11.9, −13.1, −12.5, and −10.1 Kcal/Mol, respectively. As depicted in Figure 8, lupeol bound to MMP-8 through interactions with Ala161, Gly158, and Asn218; n-docosanoic acid interacted with Ala161, Glu198, His207, Pro217, and Asn218; methyl tricosanoate interacted with Asn85, Ala161, Ala163, Gln165, His 197, Glu198, His201, and Pro217; *α*-terpinyl acetate interacted with Ala161, Val194, His 197, Glu198, His207, Pro217, and Asn218; and *α*-muurolene interacted with MMP-8 through binding with His197, Leu214, and Tyr216. In the docking of Nitric oxide reductase (iNOS), lupeol, n-docosanoic acid, methyl tricosanoate, *α*-muurolene, and gamma-sitosterol achieved the best docking scores of −13.7, −12.6, −11.8, −11.8, and −11.9 Kcal/Mol, respectively. Looking at Figure 9, we see that lupeol was able to interact with the residues of iNOS through binding with Gln478, Tyr588, Glu592, Asp597, Trp678, and Val680; n-docosanoic acid interacted with Thr324, Arg414, Cys415, Gln478, Glu592, Trp678, and Asn697; methyl tricosanoate interacted with Trp409, Arg414, Cys415, Gly417, Met570, Phe584, and Glu592; *α*-muurolene interacted with Met336, Trp678, and Val680; and gamma-sitosterol interacted with Cys415, Val677, and Trp678. In conclusion, the docking results supported and justified the biological results giving rise to a synergetic effect for all the major components of the extract.

## 3. Materials and Methods

### 3.1. Plant Material

*Tamarindus indica* L.’s (Fabaceae) various organs (Bark, leaves, seeds, and fruits) were collected from the Zoo Botanical Garden, Giza, Egypt, in October 2021. The various plant organs were kindly identified and authenticated by agricultural engineer Eng. Terase Labib, Consultant of Plant Taxonomy at the Ministry of Agriculture and El-Orman Botanical Garden, Giza, Egypt. Voucher specimens of the various plant organs with codes BUC- PHG-TIL-6, BUC- PHG-TIS-7, BUC- PHG-TIF-8, and BUC-PHG-TIB-9 were kept at the Department of Pharmacognosy, Badr University in Cairo, Cairo, Egypt.

### 3.2. Preparation of the N-Hexane Extracts of Various Organs

The dried samples of the barks, leaves, seeds, and fruits of *T. indica* (100 g each) were powdered and extracted by cold maceration method with *n*-hexane (500 mL × 3) separately, followed by filtration for 3 days. The filtrate of each plant material was completely evaporated *in vacuo* at 45 °C until dry to obtain the dried residue of the *n*-hexane extract (420 mg, 610 mg, 207 mg, and 312 mg) for barks, leaves, seeds, and fruits, respectively. The dried residue of each plant part was named as follows; *T. indica* bark (TIB), *T. indica* leaves (TIL), *T. indica* seeds (TIS), and *T. indica* fruits (TIF). All extracts were stored in a tight container at 4 °C in the refrigerator for further analysis [65].

### 3.3. GC/MS Analysis

Gas chromatography coupled with mass spectrometry (GC/MS) analyses were performed on a Shimadzu GCMS-QP 2010 (Shimadzu Corporation, Koyoto, Japan), provided using an Rtx-5MS (30 m × 0.25 mm i.d. × 0.25 µm film thickness) capillary column (Restek, Bellefonte, PA, USA) and attached to a Shimadzu mass spectrometer. The column temperature was initially set at 50 °C for 3 min. Then, the temperature was gradually increased from 50 to 300 °C at a rate of 5 °C/min and then isothermally maintained at 300 °C for 10 min. The temperature of the injector was kept at 280 °C. Helium was used as a carrier gas at a flow rate of 1.37 mL/min. The ion source and the interface were at temperatures of 280 and 220 °C, respectively. An injection of 1 µL of 1% *v*/*v* of diluted sample was achieved via a split mode adopting a split ratio of 15:1. Recording of the mass spectrum was performed in EI mode of 70 eV from m/z 35 to 500. Compound quantitation was performed based on the normalisation method, employing the reading of three chromatographic runs.

### 3.4. Identification of the N-Hexane Extract Components

The components of the *n*-hexane extracts were characterised by comparing their GC/MS spectra, fragmentation patterns, and retention indices with those reported in the literature data [66,67,68,69,70,71,72]. The retention indices were calculated relative to a homologous series of *n*-alkanes (C8–C28) injected under the same conditions.

### 3.5. Chemometric Analysis

The data obtained from GC-MS were subjected to multivariate analysis. Principal component analysis (PCA) was performed as the first step in data analysis to provide an overview of all observations and samples and to identify and evaluate groupings, trends, and strong outliers [73]. Hierarchal cluster analysis (HCA) was used to allow the clustering of samples. The clustering patterns were constructed by applying the complete linkage method. This presentation was more efficient when the distance between samples (points) was computed using the Euclidean method. Both PCA and HCA were achieved by utilising Unscrambler^®^X 10.4 from CAMO (Computer Aided Modeling, Viken, Norway) [74].

### 3.6. Anti-Inflammatory Activity

Murine macrophage RAW264.7 cells (ATCC^®^) were maintained in a complete Dulbecco’s Modified Eagle’s Medium (DMEM, Corning, NY, USA) supplemented with 10% foetal bovine serum, penicillin (100 U/mL), streptomycin sulphate (100 μg/mL), and 2 mM L-glutamine in a humidified 5% CO_2_ incubator. For passaging and treatment, cells were washed with phosphate-buffered saline and scrapped off the flasks using sterile scrappers (SPL, Spain). RAW 264.7 cell stock (0.5 × 106 cells/mL) was seeded into 96-well microwell plates and incubated overnight. The next day, the non-induced triplicate wells received a medium with the sample vehicle (DMSO, 0.1% *v*/*v*). The inflammation group of triplicate wells received the inducer of inflammation [lipopolysaccharide (LPS) as 100 ng/mL] in complete culture media containing 0.1% DMSO, *v*/*v*. Sample groups of triplicate wells received two screening amounts (10 and 50 μg/mL) of the sample dissolved in DMSO and diluted into culture media containing LPS (Final concentration of DMSO = 0.1%, by volume). L-NAME (L-N^G^-nitro arginine methyl ester) (1 mM) was used as a standard NOS inhibitor. After 24 h of incubation, a Griess assay [75] was used to determine NO in all wells. Equal volumes of culture supernatants and Griess reagent were mixed and incubated at room temperature for 10 min to form the coloured diazonium salt and read at an absorbance of 520 nm. The NO Inhibition % of the test extract was calculated relative to the LPS-induced inflammation group, normalised to cell viability determined with the Alamar Blue™ reduction assay [76].

### 3.7. Wound Healing Activity

#### 3.7.1. Cytotoxicity Assay

The cytotoxicity of the tested plant extract against the HSF (Human Skin Fibroblast) cell line was assessed prior to the wound-healing assay using the sulforhodamine B assay (SRB) [47]. The HSF cell line was obtained from Nawah Scientific Inc. (Mokatam, Cairo, Egypt). Cells were maintained in a DMEM medium supplemented with 100 mg/mL of streptomycin, 100 units/mL of penicillin, and 10% of heat-inactivated foetal bovine serum in a humidified 5% (*v*/*v*) CO_2_ atmosphere at 37 °C. Aliquots of 100 μL of cell suspension (5 × 10^3^ cells) were placed in 96-well plates and incubated in a complete medium for 24 h. Cells were treated with another aliquot of 100 μL medium containing DRSE at various concentrations (0.01, 0.1, 1, 10, and 100 μg/mL). After 72 h of drug exposure, cells were fixed by replacing the medium with 150 μL of 10% TCA and incubated at 4 °C for 1 h. The TCA solution was removed, and the cells were washed five times with distilled water. Aliquots of 70 μL SRB solution (0.4% *w*/*v*) were added and incubated in a dark place at room temperature for 10 min. Plates were washed three times with 1% acetic acid and allowed to air-dry overnight. Then, 150 μL of TRIS (10 mM) was added to dissolve the protein-bound SRB stain; the absorbance was measured at 540 nm using a BMGLABTECH^®^-FLUO star Omega microplate reader (Ortenberg, Germany).

#### 3.7.2. Scratch Wound Assay

The wound-healing activity of TIB, TIL, TIS, and TIF *n*-hexane extract was evaluated using in vitro cell migration studies on HSF cells. Cells were plated at a density of 2 × 10^5^/well onto a coated 12-well plate for the scratch wound assay and cultured overnight in 5% FBS-DMEM at 37 °C and 5% CO_2_. On the next day, horizontal scratches were introduced into the confluent monolayer; the plate was washed thoroughly with PBS, control wells were replenished with the fresh medium, and drug wells were treated with fresh media containing the drug. Images were taken using an inverted microscope at the indicated time intervals. The plate was incubated at 37 °C and 5% CO_2_ between time points. The experiment was done in triplicate. The acquired images are displayed in Figure 6 and were analysed using MII ImageView software version 3.7. Wound width was calculated as the average distance between the edges of the scratches; the wound width decreases as cell migration is induced. The results are displayed as mean ± standard deviation (Table 3) [77,78,79].

### 3.8. Statistical Analysis

Statistical analyses were done using a one-way analysis of variance (ANOVA) followed by the Tukey–Kramer multiple comparison test (*p* < 0.05). Statistical analyses were performed using GraphPad Prism 6.01 (GraphPad Inc., La Jolla, CA, USA).

### 3.9. In Silico Molecular Docking Studies

The X-ray 3D structures of glycogen synthase kinase 3-β (GSK3-β), matrix metalloproteinases-8 (MMP-8), and nitric oxide synthase (iNOS) were downloaded from the protein data bank www.pdb.org (accessed on 15 August 2022) using the following IDs: 3F88, 5H8X, and 3N2R, respectively [80,81,82]. All the docking studies were conducted using MOE 2019 [83], which was also used to generate the 2D interaction diagrams between the docked ligands and their potential targets. The identified major compounds were prepared using the default parameters and saved in a single MDB file. The active site for each target was determined from the binding of the corresponding co-crystalised ligand. Finally, the docking was finalised by docking the MDB file containing all the major compounds into the active site of the three enzymes.

## 4. Conclusions

The present study investigated the secondary metabolites in the *n*-hexane extract of *T. indica’s* various organs and their in vitro anti-inflammatory and wound healing properties using the scratch assay with HSF cells. The GC/MS analysis revealed that triterpenoids and steroids were the predominant classes in the *T. indica* bark and leaf n-hexane extracts. Additionally, the seed and fruit *n*-hexane extracts showed a high percentage of fatty acids and higher hydrocarbons. PCA and HCA successfully discriminated various *T. indica* organs based on their GC/MS metabolites. The tested extracts showed promising anti-inflammatory and wound-healing properties. Additionally, the major characterised phytoconstituents achieved promising docking scores in the active sites of glycogen synthase kinase 3-β (GSK3-β), matrix metalloproteinases-8 (MMP-8), and nitric oxide synthase (iNOS), which give the *n*-hexane extract of the various *T. indica* organs a chance to be incorporated in the pharmaceutical products for wound healing after further in vivo and clinical trials.

## Figures and Tables

**Figure 1 plants-12-00087-f001:**
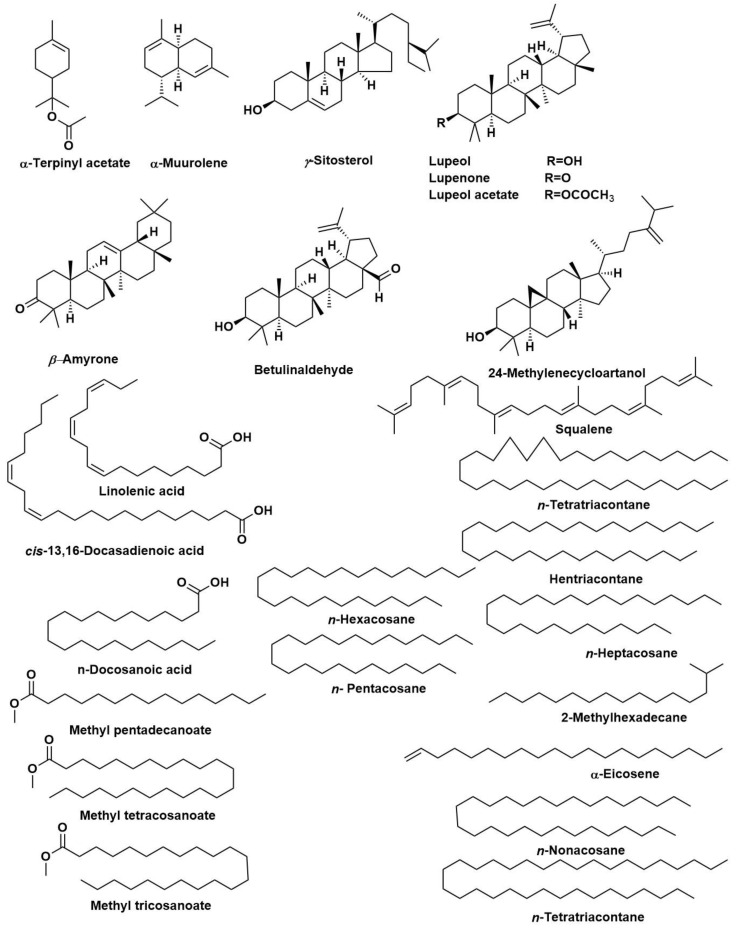
The major compounds identified in bark, leaves, seeds, and fruits of *Tamarindus indica*.

**Figure 2 plants-12-00087-f002:**
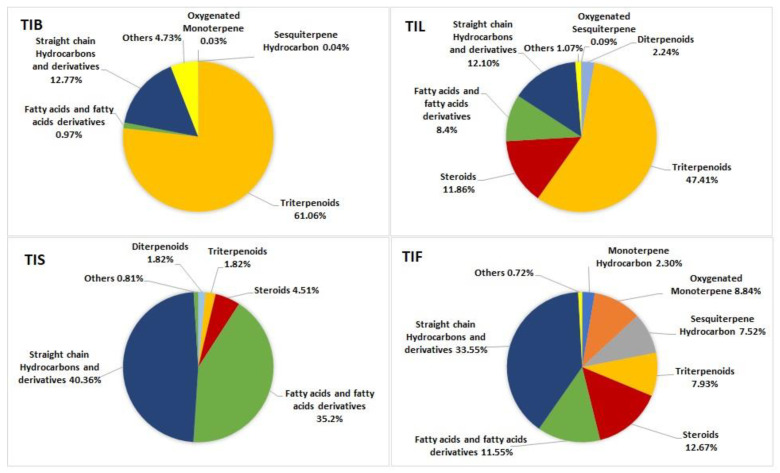
Pie charts demonstrate distribution of metabolite classes in percentages within various organs: TIB (bark), TIL (leaves), TIS (seeds), and TIF (fruits) of *Tamarindus indica*.

**Figure 3 plants-12-00087-f003:**
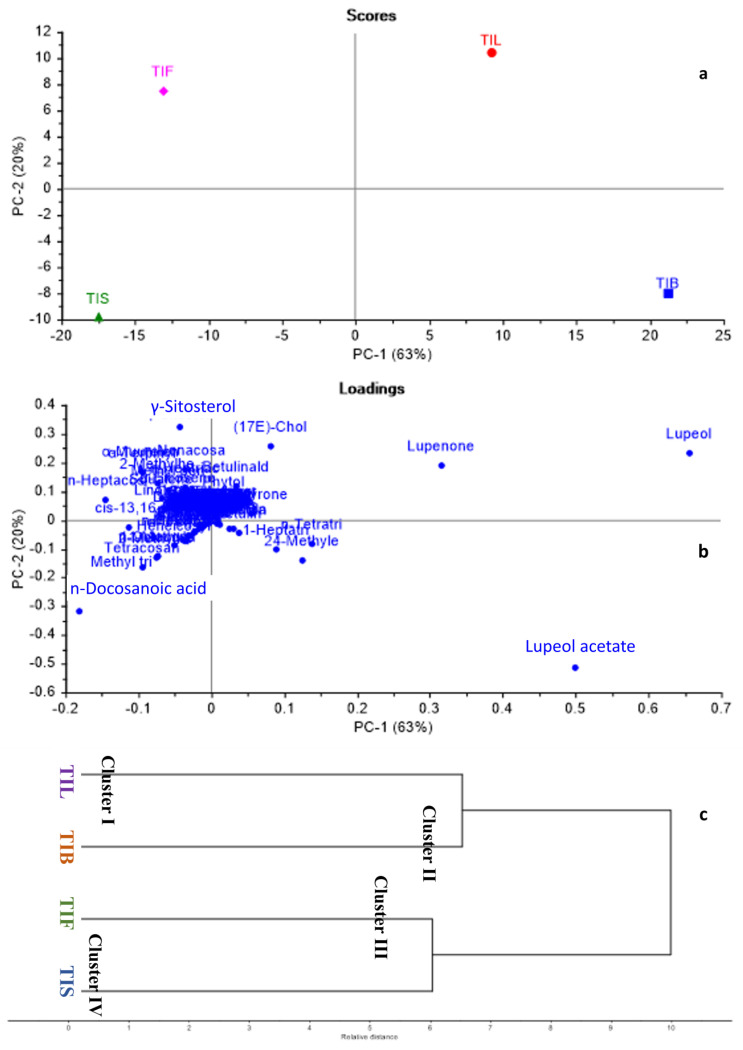
Principal component analysis score plot (**a**), loading plot (**b**), HCA dendrogram (**c**) based on GC-MS metabolites of various *T. indica* organs as identified in Table 1.

**Figure 4 plants-12-00087-f004:**
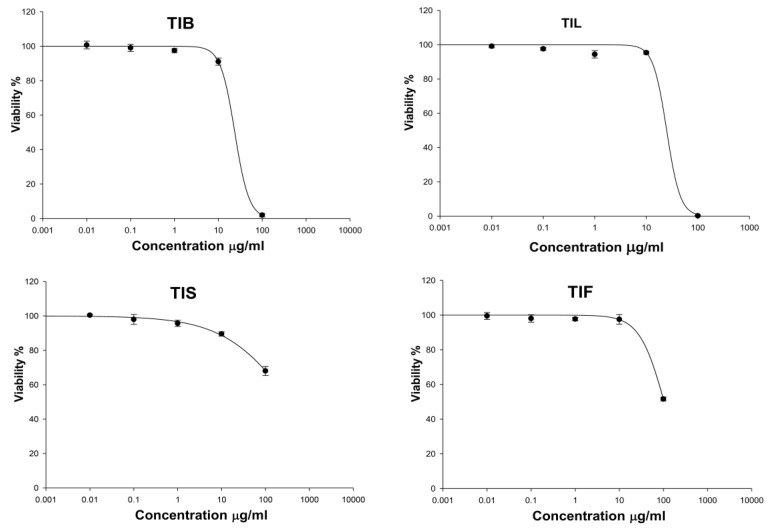
The effect of *n*-hexane extract from various organs of *T. indica*: TIB (bark), TIL (leaves), TIS (seeds), and TIF (fruits) on Human Skin Fibroblast cells (HSF) viability.

**Figure 5 plants-12-00087-f005:**
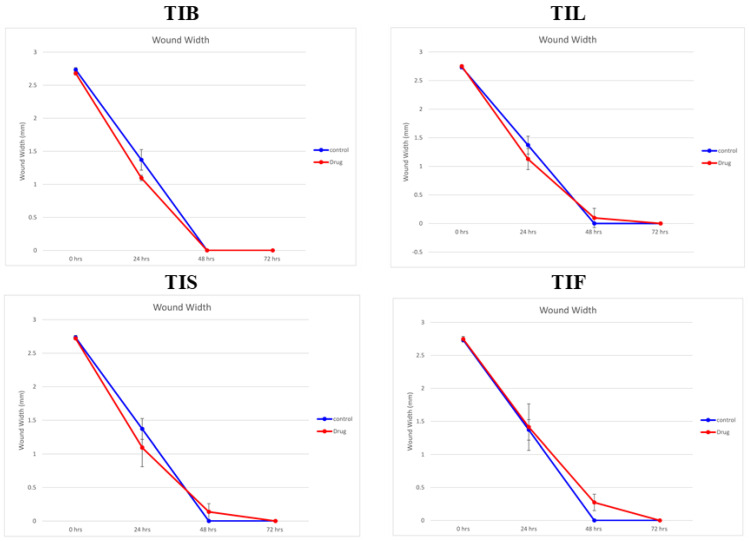
The wound width changes in the absence or presence of 10 µg/mL of *n*-hexane extract from various organs of *T. indica*: TIB (bark), TIL (leaves), TIS (seeds), and TIF (fruits).

**Figure 6 plants-12-00087-f006:**
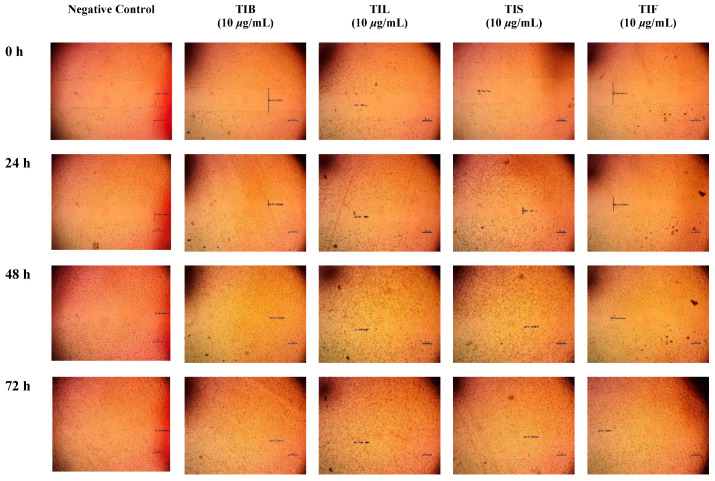
Microscopic images of (HSF) incubated in absence of the plant extract (negative control) and depicting the influence of 10 µg/mL of *n*-hexane extract from various organs of *T. indica*: TIB (bark), TIL (leaves), TIS (seeds), and TIF (fruits). Images were captured at 0, 24, 48 and 72 h. The boundaries of the scratched wounds are marked by dark lines.

**Figure 7 plants-12-00087-f007:**
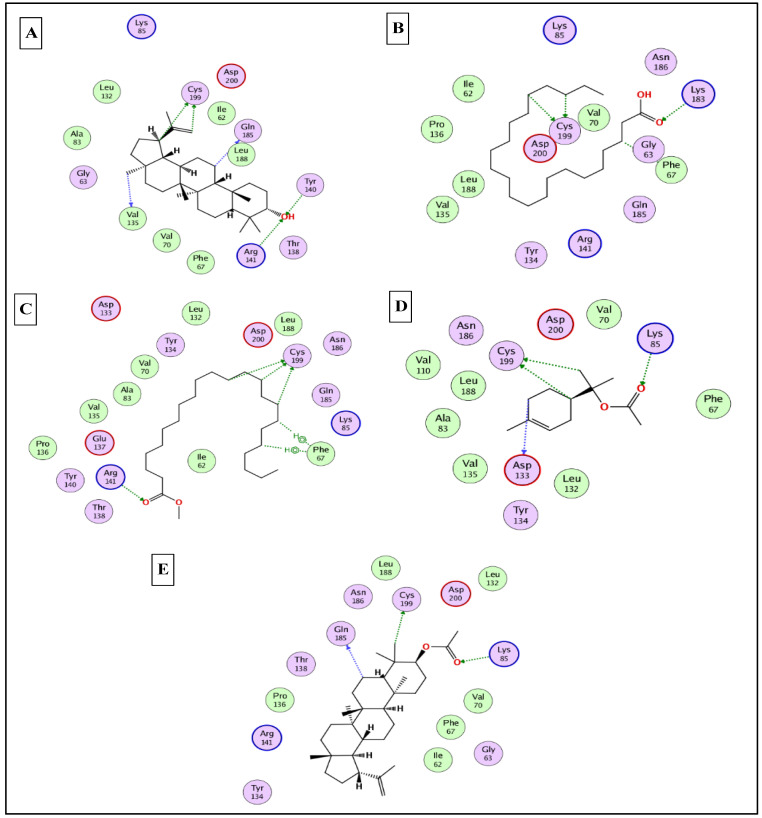
2D binding modes of lupeol (**A**), n- docosanoic acid (**B**), methyl tricosanoate (**C**), α-terpinyl acetate (**D**), and lupeol acetate (**E**) to the active binding sites of glycogen synthase kinase 3-β (GSK3-β).

**Figure 8 plants-12-00087-f008:**
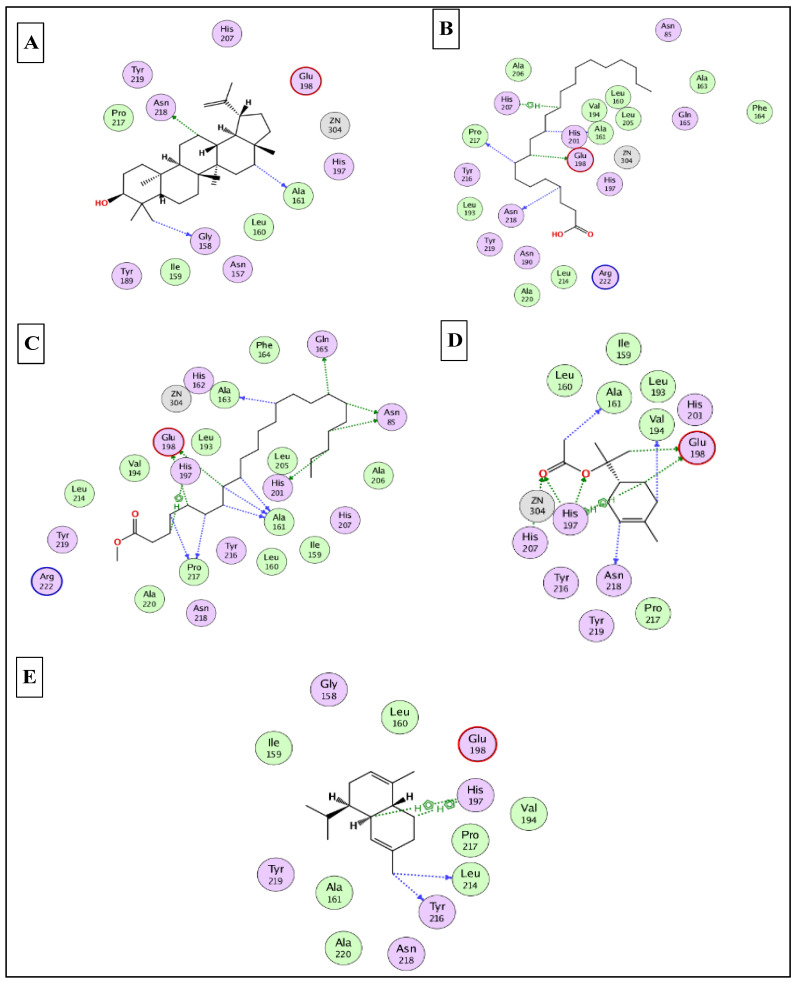
2D binding modes of lupeol (**A**), n- docosanoic acid (**B**), methyl tricosanoate (**C**), α-terpinyl acetate (**D**), and α-muurolene (**E**) to the active binding sites of matrix metalloproteinases-8 (MMP-8).

**Figure 9 plants-12-00087-f009:**
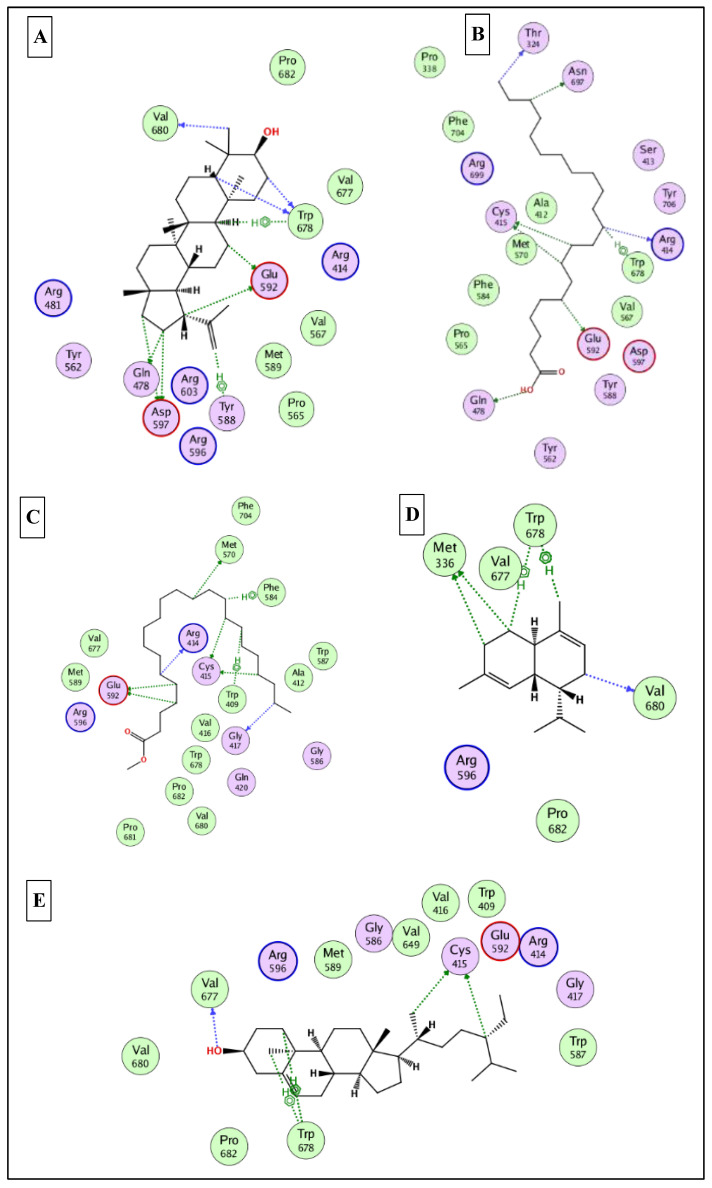
2D binding modes of (**A**), n- docosanoic acid (**B**), methyl tricosanoate (**C**), α-muurolene (**D**), and gamma-sitosterol (**E**) to the active binding sites of nitric oxide synthase (iNOS).

**Table 1 plants-12-00087-t001:** Chemical profile of *n*-hexane extract from various organs of *T. indica*: TIB (bark), TIL (leaves), TIS (seeds), and TIF (fruits) using GC/MS analysis.

No.	R_t_ _(min)_	Compound Name	RI_Exp_.^a^	RI_Lit_ ^b^	Molecular Formula	Peak Area (%)			
						TIB	TIL	TIS	TIF
1	6.94	*α*-pinene	930	930	C_10_H_16_	-	-	-	0.38
2	8.16	*α*-Sabinene	971	971	C_10_H_16_	-	-	-	0.57
3	9.90	D-Limonene	1028	1028	C_10_H_16_	-	-	-	1.35
4	9.97	1,8-Cineole	1030	1030	C_10_H_18_O	-	-	-	0.61
5	13.93	Isoborneol	1156	1156	C_10_H_18_O	-	-	-	0.87
6	19.00	*α*-Terpinyl acetate	1328	1327	C_12_H_20_O_2_	0.03	-	-	7.36
7	23.59	α-Muurolene	1494	1494	C_15_H_24_	0.04	-	-	7.52
8	27.69	2-Methylhexadecane	1668	1666	C_17_H_36_	0.03	-	-	5.70
9	31.07	Hexadecanal; Palmitaldehyde	1810	1811	C_16_H_32_O	-	-	-	0.36
10	31.25	Methyl pentadecanoate	1819	1820	C_16_H_32_O_2_	-	-	-	4.53
11	31.52	Neophytadiene	1833	1837	C_20_H_38_	-	0.2	-	-
12	31.67	Hexahydrofarnesyl acetone	1841	1842	C_18_H_36_O	-	0.09	-	-
13	32.43	Heptadecan-2-one	1880	1892	C_17_H_34_O	0.03	0.07	-	-
14	33.37	Methyl palmitate	1928	1928	C_17_H_34_O_2_	0.11	-	-	-
15	34.42	*α*-Eicosene	1980	1986	C_20_H_40_	-	-	-	3.44
16	35.62	Octadecanal	2040	2034	C_18_H_36_O	-	-	0.58	-
17	36.73	n-Heneicosane	2098	2100	C_21_H_44_	0.05	-	-	-
18	36.85	Methyl linolenate	2104	2108	C_19_H_32_O_2_	0.07	-	-	-
19	37.06	Phytol	2116	2116	C_20_H_40_O	-	1.97	-	-
20	37.32	Linolenic acid	2130	2134	C_18_H_30_O_2_	-	-	1.23	3.26
21	37.88	Ethyl linolate	2160	2164	C_20_H_36_O_2_	0.08	-	-	-
22	38.48	Stearic acid=n-Octadecanoic acid	2192	2180	C_18_H_36_O_2_	-	-	0.64	
23	38.59	n-Docosane	2198	2200	C_22_H_46_	0.07	-	0.82	-
24	38.89	Phytol acetate	2215	2218	C_22_H_42_O_2_	-	-	0.67	-
25	39.17	Methyl nonadecanoate	2231	2230	C_20_H_40_O_2_	-	-	0.59	-
26	40.12	Heneicosan-3-one	2285	2283	C_21_H_42_O	-	-	0.54	-
27	40.20	1-Eicosanol	2289	2292	C_20_H_42_O	0.09	-	-	-
28	40.36	n- Tricosane	2298	2300	C_23_H_48_	0.17	0.04	-	-
29	40.65	Methyl cis-11-eicosenoate	2315	2302	C_21_H_40_O_2_	-	-	1.83	-
30	41.15	Methyl eicosanoate	2344	2339	C_21_H_42_O_2_	-	-	0.77	-
31	41.41	4,8,12,16-Tetramethylheptadecan-4-olide	2360	2364	C_21_H_40_O_2_	-	0.07	0.40	-
32	41.57	n-Eicosanoic acid	2369	2380	C_20_H_40_O_2_	-	-	1.01	-
33	42.07	2,2′-Methylene-bis-(6-tert butyl-4-methylphenol)	2398	2398	C_23_H_32_O_2_	0.10	0.06	-	-
34	42.24	n-Tetracosane	2409	2400	C_24_H_50_	-	-	0.53	-
35	42.48	Docosanal	2424	2426	C_22_H_44_O	-	-	0.64	1.48
36	42.61	Methyl heneicosanoate	2431	2430	C_22_H_44_O_2_	0.07	-	-	-
37	42.85	1-Docosanol	2446	2456	C_22_H_46_O	-	-	0.54	-
38	43.1	2-Methyltetracosane	2461	2465	C_25_H_52_	-	-	1.28	-
39	43.48	1-Pentatricosene	2484	2485	C_35_H_70_	-	-	1.77	-
40	43.62	Cyclogallipharaol	2493	2499	C_21_H_36_O	0.35	-	0.81	0.72
41	43.70	n- Pentacosane	2498	2500	C_25_H_52_	0.24	0.23	2.46	-
42	43.87	Heneicosyl acetate	2509	2509	C_23_H_46_O_2_	-	-	2.94	-
43	43.96	Palmitic acid β-monoglyceride	2514	2519	C_19_H_38_O_4_	0.07	-	-	-
44	44.230	Methyl docosanoate	2531	2531	C_23_H_46_O_2_	0.08	-	-	-
45	44.305	11-Methylpentacosane	2536	2529	C_26_H_54_	-	-	4.22	-
46	44.675	cis-13,16-Docasadienoic acid	2560	2566	C_22_H_40_O_2_	-	-	3.70	3.76
47	44.84	n-Docosanoic acid	2570	2569	C_22_H_44_O_2_	-	-	10.49	-
48	45.18	Ethyl docosanoate	2592	2593	C_24_H_48_O_2_			1.44	
49	45.275	n-Hexacosane	2598	2600	C_26_H_54_	0.11	0.20	2.47	-
50	45.43	Methyl tricosanoate	2608	2615	C_24_H_48_O_2_	-	-	7.09	-
51	45.62	Erucylamide	2621	2625	C_22_H_43_NO	-	-	1.32	-
52	45.83	Tetracosanal	2635	2632	C_24_H_48_O	-	-	5.42	-
53	46.19	2-Methylhexacosane	2658	2662	C_27_H_56_	-	-	4.31	-
54	46.793	1-Heptacosene	2698	2694	C_27_H_54_	0.58	-	-	-
55	46.805	n-Heptacosane	2699	2700	C_27_H_56_	-	1.37	4.17	6.67
56	47.310	Methyl tetracosanoate	2733	2731	C_25_H_50_O_2_	0.14	-	2.17	-
57	47.77	2-Methylheptacosane	2765	2761	C_28_H_58_	-	-	0.59	-
58	48.02	16-Acetoxycarterochaetol	2782	2787	C_22_H_36_O_3_	-	-	0.26	-
59	48.260	n-Octacosane	2798	2800	C_28_H_58_	0.09	0.44	4.63	-
60	48.800	Squalene	2836	2835	C_30_H_50_	0.24	1.29	1.82	4.17
61	48.865	n-Hexacosanal	2841	2833	C_26_H_52_O	0.04	0.09	-	-
62	48.94	n-Hexacosanol	2846	2848	C_26_H_54_O	-	-	0.25	-
63	49.26	2-Methyloctacosane	2869	2860	C_29_H_60_	-	-	1.32	-
64	49.44	1-Nonacosene	2882	2884	C_29_H_58_	-	-	0.67	-
65	49.730	n-Nonacosane	2902	2900	C_29_H_60_	1.26	4.13	1.69	5.42
66	50.175	15-Methylnonacosane	2935	2931	C_30_H_62_	0.03	-	-	-
67	50.285	Methyl hexacosanoate	2943	2940	C_27_H_54_O_2_	-	0.17	-	-
68	50.515	2-Methylnonacosane	2960	2960	C_30_H_62_	0.04	-	-	-
69	51.040	n-Triacontane	2998	3000	C_30_H_62_	0.04	0.37	-	-
70	51.130	Benzyl icosanoate	3005	3003	C_27_H_46_O_2_	0.02	-	-	-
71	51.210	1-Heptacosanol	3011	3016	C_27_H_56_O	-	0.06	-	-
72	51.655	n-Octacosanal	3045	3039	C_28_H_56_O	-	0.29	-	0.44
73	51.850	2-Methyltriacontane	3059	3060	C_31_H_64_	-	0.09	-	-
74	52.370	n-Hentriacontane	3099	3100	C_31_H_64_	0.08	1.73	0.61	3.32
75	52.475	Octacosanol	3107	3110	C_28_H_58_O	1.02	-	-	1.89
76	52.72	13-Methylhentriacontane	3126	3130	C_32_H_66_	-	0.15	0.85	-
77	52.835	Campesterol	3134	3131	C_28_H_48_O	-	0.05	-	-
78	53.015	Cholest-3,5-diene	3148	-	C_27_H_44_	0.07	0.09	-	-
79	53.085	*α*-Tocopherol	3154	3149	C_29_H_50_O_2_	0.03	1.01	-	-
80	53.155	Ergosta-5,8,22-trien-3-ol, (3*β*,22*E*)-	3159	3158	C_28_H_44_O	-	0.40	-	-
81	53.340	Stigmasterol	3173	3170	C_29_H_48_O	0.11	0.25	-	-
82	53.475	*β*-Sitosterol	3183	3197	C_29_H_50_O	-	0.10	-	-
83	53.710	n-Dotriacontane	3201	3200	C_32_H_66_	0.26	0.23	-	-
84	53.925	Cholest-5-ene, 3*β*-methoxy	3216	3216	C_28_H_48_O	-	0.08	-	-
85	54.375	Triacontanal	3248	3251	C_30_H_60_O	-	0.47	-	0.69
86	54.575	5*α*-Stigmast-22-en-3β-ol	3261	3253	C_29_H_50_O	1.09	0.84	0.72	1.39
87	55.070	Chondrillasterol	3296	3295	C_29_H_48_O	1.42	0.07	-	-
88	55.130	n-Tritriacontane	3300	3300	C_33_H_68_	-	0.41	-	-
89	55.230	Lanosterol	3306	3302	C_30_H_50_O	0.25	0.13	-	-
90	55.310	1-Triacontanol	3311	3306	C_33_H_68_O	0.38	-	-	-
91	55.440	5*α*-Stigmastan-3*β*-ol	3320	3325	C_29_H_52_O	0.04	0.14	-	-
92	55.655	*β*-Amyrin	3332	3337	C_30_H_50_O	0.11	-	-	-
93	56.015	γ-Sitosterol	3353	3351	C_29_H_50_O	4.17	8.83	3.79	11.28
94	56.200	2-Methyldotriacontane	3364	3259	C_33_H_68_	-	0.52	-	0.76
95	56.335	*β*-Amyrone	3372	3327	C_30_H_50_O	1.78	2.05	-	-
96	56.445	*α*-Amyrin	3379	3376	C_30_H_50_O	-	0.33	-	-
97	56.740	n-Tetratriacontane	3397	3403	C_34_H_70_	8.16	1.21	-	3.38
98	57.020	*β*-Amyrin acetate	3446	3437	C_32_H_52_O_2_	-	0.08	-	-
99	57.195	Lupenone	3488	3384	C_30_H_48_O	9.36	13.08	-	-
100	57.550	Lupeol acetate	3518	3525	C_32_H_52_O_2_	23.02	1.68	-	-
101	57.68	Lupeol	3527	3500	C_30_H_50_O	23.61	22.78	-	3.76
102	58.015	(17*E*)-Cholesta-17,24-diene-3,6-diol	3550	-	C_27_H_44_O_2_	0.29	8.23	-	-
103	58.170	Hexadecanoic acid, 3,7,11,15-tetramethyl-2-hexadecenyl ester	3561	3567	C_36_H_70_O_2_	0.04	-	-	-
104	58.295	24-Methylenecycloartan-3-one	3570	-	C_31_H_50_O	0.45	-	-	-
105	58.605	24-Methylenecycloartanol	3591	-	C_31_H_52_O	5.87	0.18	-	-
106	58.890	3*β*-Hydroxystigmast-5-en-7-one	3611	3609	C_29_H_48_O_2_	-	1.03	-	-
107	59.095	Betulinaldehyde	3625	3628	C_30_H_48_O_2_	-	3.70	-	-
108	59.440	Germanicol	3649	-	C_30_H_50_O	-	0.80	-	-
109	61.000	Betulin	3757	3760	C_30_H_50_O_2_	1.78	-	-	-
110	62.04	Ursane-3,12-diol	3829	-	C_30_H_52_O_2_	-	0.59	-	-
111	62.11	1-Heptatriacontanol	3834	-	C_37_H_76_O	4.25	-	-	-
112	62.39	Stigmastane-3,6-dione	3853	3601	C_29_H_48_O_2_	1.16	-	-	-
113	62.79	9,19-Cyclolanost-23-ene-3,25-diol, (3*β*,23*E*)-	3881	-	C_32_H_52_O_3_	-	0.70	-	-
		Monoterpene Hydrocarbon				-	-	-	2.30
		Oxygenated Monoterpene				0.03	-	-	8.84
		Sesquiterpene Hydrocarbon				0.04	-	-	7.52
		Oxygenated Sesquiterpene				-	0.09	-	-
		Diterpenoids				-	2.24	1.33	-
		Triterpenoids				61.06	47.41	1.82	7.93
		Steroids				13.47	11.86	4.51	12.67
		Fatty acids and fatty acids derivatives				0.97	8.40	35.22	11.55
		Straight-chain Hydrocarbons and derivatives				12.77	12.1	40.36	33.55
		Others				4.73	1.07	0.81	0.72
		Total identified compounds %				93.07	83.17	84.05	85.08

Compounds listed in order of their elution in RTX-5 GC column. Identification was based on comparison of the compound mass spectral data (MS) and retention indices (RI) with those of NIST Mass Spectral Library (2011), Wiley Registry of Mass Spectral Data 8th edition and literature. ^a^ Retention index calculated experimentally in RTX-5 column relative to n-alkane series (C8–C28). ^b^ Published retention indices.

**Table 2 plants-12-00087-t002:** The anti-inflammatory effects of the *n*-hexane extract from various organs of *T. indica*: TIB (bark), TIL (leaves), TIS (seeds), and TIF (fruits) on lipopolysaccharide (LPS)-induced RAW 264.7 macrophages.

Sample	%NO Inhibition	
	10 μg/mL	100 μg/mL
TIB	7.53 ±1.69 ^b^	51.44 ± 1.17 ^b^
TIL	53.97 ± 5.89 ^b^	98.00 ± 1.90 ^b^
TIS	19.54 ± 1.19 ^b^	85.47 ± 0.22 ^a^
TIF	26.66 ± 3.44 ^b^	83.69 ± 2.39 ^a^
L-NAME (1 mM)	84.64 ± 1.04 ^a^	

Means bearing same scripts (^a^ or ^b^) are not significantly different from control at *p* < 0.05, Mean ± Standard error.

**Table 3 plants-12-00087-t003:** Wound width of the scratched Human Skin Fibroblast cells (HSF) incubated in the absence of the plant extract (negative control) and the presence of *n*-hexane extract from *T. indica*: TIB (bark), TIL (leaves), TIS (seeds), and TIF (fruits) (10 μg/mL).

Time (h)	Wound Width (mm)				
	TIB (10 μg/mL)	TIL (10 μg/mL)	TIS (10 μg/mL)	TIF (10 μg/mL)	Control
0	2.68 ± 0.02 ^a^	2.75 ± 0.02 ^a^	2.72 ± 0 ^a^	2.74 ± 0.04 ^a^	2.73 ± 0.03 ^a^
24	1.09 ± 0.04 ^a^	1.12 ± 0.18 ^a^	1.09 ± 0.28 ^a^	1.41 ± 0.35 ^a^	1.37 ± 0.15 ^a^
48	0 ^a^	0.09 ± 0.16 ^a^	0.13 ± 0.12 ^a^	0.27 ± 0.12 ^a^	0 ^a^
72	0	0	0	0	0

Means bearing same script (^a^) are not significantly different from control at *p* < 0.05, Mean ± Standard error.

**Table 4 plants-12-00087-t004:** The docking scores achieved by the major identified compounds against the three enzymes.

Compound Name	Docking Scores (Kcal/mol)		
	Glycogen Synthase Kinase 3-β (GSK3-β) 3F88	Nitric Oxide Reductase (iNOS) 3N2R	Matrix Metalloproteinases 8 (MMP-8) 5H8X
Lupeol	−12.5	−13.7	−9.8
n-docosanoic acid	−11.7	−12.6	−11.9
methyl tricosanoate	−11.2	−11.8	−13.1
*α*-terpinyl acetate	−11.8	−10.1	−12.5
*α*-muurolene	−10.4	−11.8	−10.1
Gamma- sitosterol	−10.2	−11.9	−8.2
Lupenone	−10.1	−10.3	−7.6
Lupeol acetate	−11.3	−9.5	−7.7
n-tetratriacontane	−9.8	−7.6	−7.5
Betulinaldehyde	−8.7	−8.1	−7.4
*β*-Amyrone	−9.2	−6.5	−7.1
24-methylenecycloartanol	−8.3	−7.2	−6.8
methyl tricosanoate	−7.8	−7.4	−7.5
n-hexacosane	−8.2	−7.8	−7.3
*cis*-13,16-docasadienoic acid	−7.5	−8.3	−6.5
n-tetratriacontane	−7.9	−9.1	−5.9
methyl pentadecanoate	−7.3	−7.8	−7.6
Squalene	−6.8	−7.8	−7.2
Linolenic acid	−8.1	−8.5	−7.9
n- pentacosane	−7.3	−7.7	−7.4

## Data Availability

Data supporting the reported results can be found at NIST Chemistry Webbook, https://webbook.nist.gov/chemistry/ (accessed on 10 October 2022).

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
