# Peer review of "Comparative Metabolic Study of Tamarindus indica L.’s Various Organs Based on GC/MS Analysis, In Silico and In Vitro Anti-Inflammatory and Wound Healing Activities"

_plants, 2022, doi:10.3390/plants12010087_

Round 1
Reviewer 1 Report
Dear Authors,
Despite all advantages (actuality, novelty, scientific soundness, proper presentation of obtained data etc.) of the research performed, the manuscript contains several minor shortcomings, and they should be eliminated before acceptance:
In Introduction or discussion section should be included info about using of remedies/preparations made from Tamarindus indica in folk medicine/ethnopharmacology of various nations for anti‑inflammatory and wound healing purposes.
Line 383: have to be extracts in n-hexane, but not essential oils
Author Response
Response to Reviewers comments
Firstly, we would like to thank you for the constructive comments concerning our article (Plants- 2109624). These comments are valuable and helpful for improving the article. All the authors have seriously discussed the reviewer’s comments, we have tried the best to modify our manuscript to meet with the requirements.
Reviewer 1:
Despite all advantages (actuality, novelty, scientific soundness, proper presentation of obtained data etc.) of the research performed, the manuscript contains several minor shortcomings, and they should be eliminated before acceptance:
Response: We are grateful for the reviewer valuable comments and suggestions. All comments were addressed point-by-point in the revised manuscript.
- In the Introduction or discussion section should be included info about using of remedies/preparations made fromTamarindus indica in folk medicine/ethnopharmacology of various nations for anti‑inflammatory and wound healing purposes.
Response: Information about using Tamarindus indica in folk medicine had been added in the introduction part and highlighted in yellow. Page 2. Line 58-65
- Line 383: have to be extracts in n-hexane, but not essential oils
Response: It had been revised and updated as required (line 377)

Reviewer 2 Report
Article
Comparative Metabolic Study of Tamarindus indica L. Different Parts Based on GC/MS Analysis, in silico and in vitro Anti‑inflammatory and Wound Healing Activities
A brief summary
The idea of searching for the properties of these plant organs, which are usually waste, and finding potential uses for them is now often explored, and this is a very good direction. In this paper the composition of Tamarindus indica bark, leaves, seeds and fruits n-hexane extract were investigated using GC/MS as well as in silico and in vitro anti-inflammatory and wound healing activities described in details. The title in general fit the work range and presented results. The research is well designed and performed but documentation and description needs some fixes and additions. The paper is almost properly constructed. The style of presentation is quite appropriate for scientific journal.
Broad comments
1. Is the term 'different plant parts' appropriate in the context of a scientific article. (Specifically published in the Plants journal.) It appears in the title of the article itself and very often (perhaps even overused) in the paper's content. The Authors themselves use the term 'various plant organs' (lines 178-179) and this is far more appropriate. On the other hand, how about using the term "different plant raw materials" in the context of future applications?
2. The review/introduction lacks information on which compounds the Authors studied and why, and thus also why they used n-hexane as an extraction solvent. Such information is detailed in the Results (lines 214-215, 258-292) but without direct reference to the results obtained by the Authors, and should therefore be moved to the Introduction.
3. How precisely was the extraction of the plant material - a key step for the composition of the extracts studied subsequently - carried out? What was the amount of solvent, its temperature, what type of extraction was used, how long did it last? (lines 362-363)
4. The question would also be whether it makes sense to compare the results obtained for hexane extracts with methanol extracts (line 141) or those obtained by decoction (so probably aqueous) (line 203).
5. Since only a percentage of the peak area was obtained by GC, talking about concentration (line 99) is not quite appropriate.
6. The same description of the test samples should be used throughout the paper, e.g. as in Table 1 OR as in Figure 5.
7. The Authors seem to have become very involved in emphasizing the importance of their research in several places and have used perhaps too emotional means of expression for a scientific article. See lines 98, 117, 124, 157, 164 and 317.
Specific comments
Line 90 What is the yield in question?
Lines 175-176 The sentence makes no sense. These are the same results just presented differently, so they cannot endorse themselves.
Table 2 and Figure 4. The table and graph contain the same data, with completely different standard deviations. The lack of statistical analysis does not allow a reliable evaluation of the results.
Table 3 The lack of statistical analysis does not allow a reliable evaluation of the results.
Figure 5 and 6 The figures are unreadable, especially 6.
Author Response
Response to Reviewers comments
Firstly, we would like to thank you for the constructive comments concerning our article (Plants- 2109624). These comments are valuable and helpful for improving the article. All the authors have seriously discussed the reviewer’s comments, we have tried the best to modify our manuscript to meet with the requirements.
Reviewer 2:
A brief summary
The idea of searching for the properties of these plant organs, which are usually waste, and finding potential uses for them is now often explored, and this is a very good direction. In this paper the composition of Tamarindus indica bark, leaves, seeds and fruits n-hexane extract were investigated using GC/MS as well as in silico and in vitro anti-inflammatory and wound healing activities described in details. The title in general fit the work range and presented results. The research is well designed and performed but documentation and description needs some fixes and additions. The paper is almost properly constructed. The style of presentation is quite appropriate for scientific journal.
Response: We are grateful for the reviewer valuable comments and suggestions. All comments were addressed point-by-point in the revised manuscript.
Broad comments
- Is the term 'different plant parts' appropriate in the context of a scientific article. (Specifically published in the Plants journal.) It appears in the title of the article itself and very often (perhaps even overused) in the paper's content. The Authors themselves use the term 'various plant organs' (lines 178-179) and this is far more appropriate. On the other hand, how about using the term "different plant raw materials" in the context of future applications?
Response: 'different plant parts' had been replaced with 'various plant organs' in all the MS.
Using the term (different plant raw materials) will be considered in future publications.
- The review/introduction lacks information on which compounds the Authors studied and why, and thus also why they used n-hexane as an extraction solvent. Such information is detailed in the Results (lines 214-215, 258-292) but without direct reference to the results obtained by the Authors and should therefore be moved to the Introduction.
Response: Most of the essential oils were extracted by n-hexane as reported in previous literature, so references of using the n-hexane as extraction solvent was added in the introduction part. Page 2 line 81-83.
Regarding mentioned lines, lines 214-215, 258-292, the authors compare their results with that reported in literature for the main compounds with anti-inflammatory and wound healing properties with rephrasing of all the sentences to support our study.
- How precisely was the extraction of the plant material - a key step for the composition of the extracts studied subsequently - carried out? What was the amount of solvent, its temperature, what type of extraction was used, how long did it last? (Lines 362-363)
Response: The required details had been added in the 3.2. section. Page 20 line 355-357.
- The question would also be whether it makes sense to compare the results obtained for hexane extracts with methanol extracts (line 141) or those obtained by decoction (so probably aqueous) (line 203).
Response: sentence was deleted
Line 206 (decoction of the bark…..) the authors report the use of the bark in folk medicine as anti-inflammatory agent.
Response: sentence was deleted
- Since only a percentage of the peak area was obtained by GC, talking about concentration (line 99) is not quite appropriate.
Response: (Concentrations) had been replaced with (percentage). Page 3 line 107.
- The same description of the test samples should be used throughout the paper, e.g. as in Table 1 OR as in Figure 5.
Response: The same description of the samples had been updated as required throughout the paper.
- The Authors seem to have become very involved in emphasizing the importance of their research in several places and have used perhaps too emotional means of expression for a scientific article. See lines 98, 117, 124, 157, 164 and 317.
Response: All sentences were modified and rephrashed.
Specific comments
- Line 90 What is the yield in question?
Response: It had been clarified and updated as required; The yields of extraction of TIB, TIL, TIS, and TIL using n-hexane. Page 3 line 98.
- Lines 175-176 The sentence makes no sense. These are the same results just presented differently, so they cannot endorse themselves.
Response: the sentence was deleted.
- Table 2 and Figure 4. The table and graph contain the same data, with completely different standard deviations. The lack of statistical analysis does not allow a reliable evaluation of the results.
Response: Figure 4 had been removed, and statistical analysis had been added.
- Table 3 The lack of statistical analysis does not allow a reliable evaluation of the results.
Response: Statistical analysis had been added as required.
- Figure 5 and 6 The figures are unreadable, especially 6.
Response: All Figures in the Manuscript had been improved and updated.
